# The mobility of packaged phage genome controls ejection dynamics

Alex Evilevitch[1,2]*

[1]Department of Pathobiology, Division of Microbiology and Immunology, College of Veterinary Medicine, University of Illinois at Urbana-Champaign, Champaign, United States; [2]Department of Experimental Medical Sciences, Virus Biophysics Group, Lund University, Lund, Sweden

**Abstract** The cell decision between lytic and lysogenic infection is strongly influenced by dynamics of DNA injection into a cell from a phage population, as phages compete for limited resources and progeny. However, what controls the timing of viral DNA ejection events was not understood. This in vitro study reveals that DNA ejection dynamics for phages can be synchronized (occurring within seconds) or desynchronized (displaying minutes-long delays in initiation) based on mobility of encapsidated DNA, which in turn is regulated by environmental factors, such as temperature and extra-cellular ionic conditions. This mechano-regulation of ejection dynamics is suggested to influence viral replication where the cell's decision between lytic and latent infection is associated with synchronized or desynchronized delayed ejection events from phage population adsorbed to a cell. Our findings are of significant importance for understanding regulatory mechanisms of latency in phage and Herpesviruses, where encapsidated DNA undergoes a similar mechanical transition.

DOI: https://doi.org/10.7554/eLife.37345.001

## Introduction

Gaining insight into the dynamics of viral gene delivery into a host cell during infection is critical for understanding the virus replication dynamics (*Ellis and Delbrück, 1939*; *You and Yin, 1999*; *Gallet et al., 2011*; *Lee et al., 1997*) and viral fitness (*Burch and Chao, 2000*; *Abedon and Culler, 2007*). A central method for studying virus replication dynamics is the 'one-step growth cycle' method, which was developed for an ex vivo model of phage growth in *E. coli* by Ellis and Delbruck (*Ellis and Delbrück, 1939*). This quantitative method measures the rate of one cycle of phage replication. For accurate quantification of the infectious cycle rate, it is critical that the infection is synchronized so that cells are infected at the same time. This is achieved by incubating bacterial culture with phage, diluting the solution after phage has adsorbed to cells so that no more adsorption occurs, and then measuring virus growth at different time intervals using a plaque assay. Modified versions of the one-step and multi-step replication assays remain central today for monitoring the growth and evolution of viruses as they mutate to evade antiviral drugs, the immune system or environmental factors inhibiting viral spread (*Lee et al., 1997*; *Coffin, 1995*; *Perelson et al., 1996*; *Nowak, 1992*). The one-step growth cycle method builds on an assumption that initiation of infection through phage DNA injection into cells occurs rapidly and simultaneously after phage adsorbs to cells (*Ellis and Delbrück, 1939*). Although the dynamics of the phage adsorption step has been investigated (*Moldovan et al., 2007*; *Mackay and Bode, 1976*), the dynamics of the initiation of phage injection is poorly understood even though it has critical importance for virus growth rate studies.

The one-step growth cycle method focuses on the lytic infection pathway (where a virus replicates immediately, leading to cell lysis). However, the cell decision between a lytic or lysogenic infection

*For correspondence:
alexe@illinois.edu

Competing interests: The author declares that no competing interests exist.

**eLife digest** Viruses are tiny 'parasites' that smuggle their genetic material inside a cell and then hijack its resources for their own benefit. A viral infection can either be lytic or latent. In a lytic cycle, viruses make their host produce many copies of themselves, ultimately killing the cell. In contrast, during a latent infection, the viruses go 'dormant': for instance, some of them can insert their genetic material into the DNA of their host, which then gets passed on as the cell divides.

Certain viruses are capable of both lytic and latent infections. One example is the lambda phage, which targets *Escherichia coli* bacteria. In the first stage of infection, the genetic material 'shoots out' of the virus and gets injected inside the bacterium. The dynamics of the ejection process determine the type of infection that will follow. If multiple phages release their genomes quickly and within seconds of each other into the same cell, the bacterium tends to incorporate the viral DNA into its own genome, leading to a latent cycle. If the infections take place more slowly and not all at the same time, the cell is more likely to go through a lytic phase. However, the mechanism behind the different injection behaviors is still unknown; in particular, it is unclear which factors control the specificities of the ejection process in the first place.

Here, Alex Evilevitch demonstrates that the mechanical state of the phage DNA just before ejection dictates how the genetic material will then be injected in the bacteria. The experiments measured the stiffness of the DNA and the amount of heat given off during infection. Like fluid toothpaste, if the DNA is more liquid and flexible, it gets ejected quickly and simultaneously from several phages. Then, the genetic information of these viruses can be incorporated in the genome of the bacteria. On the other hand, if the DNA is more solid, it is likely to 'stick' and take time before it can be squeezed out: the injections become unsynchronised, which leads to a lytic phase.

Evilevitch then shows that the environment can influence the properties of the phages' genome. A little more heat, or certain chemicals, can make the DNA more fluid inside the viruses, and change the way it can be injected inside the bacteria.

Many viruses that cause diseases in humans – from cold sores to glandular fever – can switch between the lytic and latent cycles. For the first time, these results show that the mechanical properties of the DNA inside a virus influence the 'decision' between the two types of infection. This knowledge could help us prevent infections from becoming lytic and ultimately allow us to control the spread of disease.

DOI: https://doi.org/10.7554/eLife.37345.002

(latent state where phage DNA is integrated into a cell chromosome without killing the cell) may depend on the dynamics of DNA injection from multiple phages adsorbed to a single bacterial cell (*Zeng et al., 2010*; *Trinh et al., 2017*; *Kourilsky, 1974*; *Kourilsky, 1973*). Some studies claimed that DNA ejection from such a phage population occurs in a stochastic manner and is solely attributed to the opening dynamics of the portal-tail connector channel through which DNA is being ejected (*Raspaud et al., 2007*). This description fits the generalized picture of biochemical stochasticity proposed in the past for many cellular processes (*Arkin et al., 1998*; *Singh and Weinberger, 2009*). However, as shown below, our data does not support these claims. Furthermore, biochemical stochasticity may not play as much of a role in virus-cell interactions as previously thought (*Zeng et al., 2010*; *St-Pierre and Endy, 2008*), specifically concerning the process of cell-fate decision making between lysogenic and lytic pathways in *E. coli* following phage λ infection (*Zeng et al., 2010*; *Trinh et al., 2017*). While host functions can influence this decision process (*Casjens and Hendrix, 2015*), the initial choice is made at the individual phage level (*Trinh et al., 2017*). This, in turn, determines how phages interact with one another during cell infection, where the central interaction parameter is replication dynamics of multiple viral genomes being injected into a cell. Interestingly, replication dynamics occurs on the timescale of the genome injection process, varying from seconds to minutes (*Mackay and Bode, 1976*; *Van Valen et al., 2012*). This is also evidenced by an important role that the order of phage genome injection is playing in CRISPR sequence acquisition (*Modell et al., 2017*). Therefore, timing of phage DNA injections into a cell has been shown to affect the decision between lysogeny and cell lysis (*Trinh et al., 2017*; *Cortes et al., 2017*). Specifically, it was proposed that simultaneous, instant injection events from all

phages infecting one cell lead to a lack of competition for replication of phage genomes, which results in higher lysogenizaton frequency (*Trinh et al., 2017*; *Kourilsky, 1973*). It was indeed observed that probability of lysogeny increases with the number of simultaneously infecting phages (*Trinh et al., 2017*; *Kourilsky, 1973*). On the contrary, delayed asynchronous injections lead to competitive interactions between phage genomes observed during lytic infection. Consistently, infection delays were shown to decrease the chance of lysogeny by also lowering CII levels in the cell, where CII is a viral DNA-binding transcription factor governing postinfection decision-making (*Cortes et al., 2017*). Lytic phage that delivers its genome first overrides cell's lysogeny decision and uses cell's limited resources to replicate its own progeny (*Trinh et al., 2017*; *Kourilsky, 1973*).

A pioneering study demonstrated that DNA injection from phage λ, after preadsorption to *E. coli*, occurred instantaneously at 37°C, but at lower temperatures the heterogeneous injection dynamics with minutes-long lag times was observed, with injections completely inhibited at 4° C (*Mackay and Bode, 1976*). The mechanism behind this injection behavior remained unknown. Thus, understanding the mechanism and environmental factors determining the switch between synchronized and desynchronized injection dynamics from a phage population is of key importance for studies of the infectious cycle where the cell decision between lytic and lysogenic infection plays a crucial role. As shown here, the main factor influencing injection dynamics is the structural state of DNA inside the viral capsid. DNA is tightly packaged inside phage λ capsid with DNA-DNA surface separation of only 7 Å (*Lander et al., 2013*; *Earnshaw and Harrison, 1977*). This leads to high interstrand repulsive forces that, along with DNA bending stress, result in internal genome pressure of tens of atmospheres (*Evilevitch et al., 2003*; *Tzlil et al., 2003*; *Purohit et al., 2003*). This pressure powers DNA ejection into a bacterial cell. However, the small interstrand distance also generates electrostatic and hydration sliding friction between neighboring DNA strands (*Gabashvili and Grosberg, 1992*; *Berndsen et al., 2014*; *Liu et al., 2014*), which limits the packaged genome's mobility (or fluidity). Using atomic force microscopy (AFM) combined with isothermal titration calorimetry (ITC), we have recently shown that restricted intra-capsid DNA mobility can be characterized by a solid-like mechanical response to nano-indentation of a DNA-filled capsid (*Liu et al., 2014*; *Sae-Ueng et al., 2014*). At the same time, DNA packaged in the capsid can undergo a mechano-structural transition from a solid- to a fluid-like state. This transition is induced by increasing the temperature (which affects DNA bending stress and packing defects) or by varying external ionic conditions (which affects DNA-DNA repulsive interactions and overall genome stress in the capsid) (*Li et al., 2015*). In the fluid-like state, the interstrand DNA-DNA friction of the encapsidated genome is significantly reduced due to an increase in distance between packaged DNA strands occurring with local DNA disordering. These observations suggest that dynamics of DNA ejection can be affected by a transition in mobility of the encapsidated genome. To investigate the effect of this mechano-structural DNA transition on DNA ejection dynamics from phage λ, we designed a new ITC assay that provides ultra-sensitive detection of ejection dynamics from a phage population triggered by instant mixing with LamB λ-receptor in vitro. We also used time-resolved small angle X-ray scattering (SAXS) to further support our findings. We discovered striking bimodal dynamics of coexisting phage populations displaying fast synchronized and slow desynchronized ejection events. Here, we show that this complex ejection dynamics is directly linked to the two mechano-structural states of DNA—solid- and fluid-like—inside the λ capsid.

This remarkable observation of mechano-regulation of phage genome ejection dynamics can be compared to gene expression regulation in meters-long, tightly packaged chromosomal DNA in eukaryotic cells (*Eslami-Mossallam et al., 2016*), where gene expression is controlled not only by genetic regulation but also by mechanical properties of DNA in nucleosomes. Environmental stress factors, such as an abrupt change in temperature or in ionic conditions, were demonstrated to play a major role in the mechano-regulated gene activation process, turning certain genes on and off, in a process termed epigenetics. Similarly, we show that temperature and ionic conditions control the mechanical properties of virally encapsidated DNA and act as an on-off switch between synchronized and desynchronized ejection dynamics. This can influence timing of viral replication and the lytic-lysogenic cell decision process (*Trinh et al., 2017*). These in vitro ejection dyamics results are in agreement with earlier in vivo observations of temperature's effect on DNA injection dynamics from phage λ preadsorbed to *E. coli* (*Mackay and Bode, 1976*).

## Results and discussion

### Dynamics of DNA ejection from phage measured by ITC

Dynamics of DNA ejection from phage has been previously investigated using single particle fluorescence (*Grayson et al., 2007*) as well as bulk light scattering analysis (LS) (*Raspaud et al., 2007*; *Freeman et al., 2016*). Single particle fluorescence studies have been mainly focused on determining DNA translocation rates from a phage capsid, rather than population ejection dynamics, due to limited statistics as well as uneven timing for delivery of phage receptors to trigger ejection (because of flow cell geometry). LS measurements, on the other hand, record a decrease in total scattering intensity, *I(0)*, from a phage population. This decrease is associated with a change in DNA scattering density inside capsids, reflecting the number of DNA-filled phage particles remaining as ejection events occur. When DNA is ejected and relaxed in solution to a micron-size coil, it no longer contributes to the scattering intensity (*Freeman et al., 2016*). *Figure 1* shows our LS-measured decay of normalized scattering intensity versus time, $\Delta I(t) = \frac{I(t) - I(final)}{I(initial) - I(final)}$, for DNA ejection from wild-type (wt) DNA length phage λ (corresponding to 48,500 bp) when it is mixed with LamB

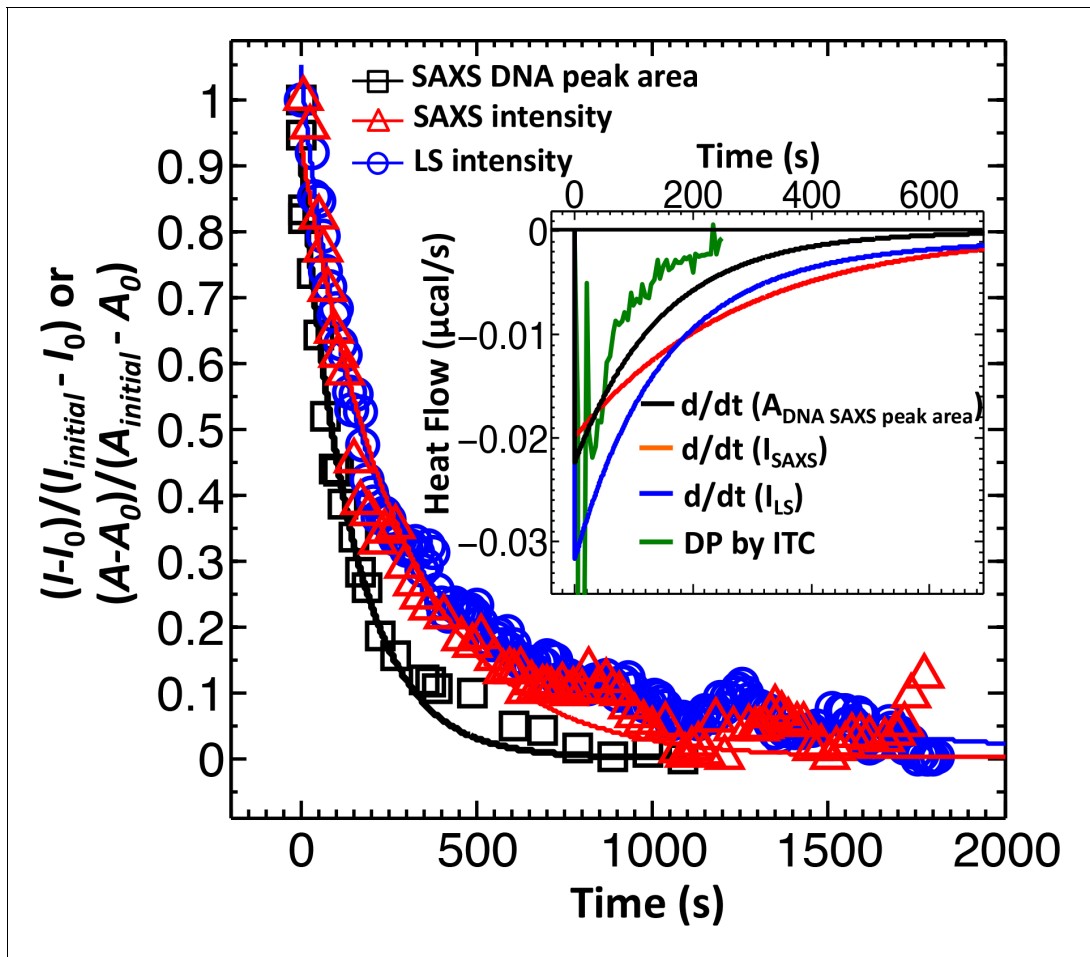

**Figure 1.** Direct comparison between LS, SAXS and ITC data for phage λ DNA ejection dynamics. The derivative with respect to time of the LS-measured normalized forward scattering curve yields the instantaneous change in the number of ejecting phages versus time ($d\Delta I/dt$). This resulting $d\Delta I/dt(t)$ curve is scaled to correspond to the ITC measured enthalpy released from a phage population during DNA ejection process. Figure shows comparison of DNA ejection dynamics for phage λ mixed with LamB receptor at 22°C measured by LS [*I(0)* versus time], time-resolved SAXS [*I(0)* versus time], and time-resolved SAXS [DNA diffraction peak area versus time]. [SAXS data is discussed in the Supplemental Meterial Section.] The lines connecting the data points are exponential fits for each data set. Figure inset shows the ITC titration DP curve (green) compared to derivatives of LS and SAXS data, which have been scaled to match the phage concentration in an ITC measurement.
DOI: https://doi.org/10.7554/eLife.37345.003

solution at 22°C in TM-buffer (10 mM $MgCl_2$, 50 mM TrisHCl, pH 7.5). As observed in the figure, the overall time for the ejection process is much slower (here > 1000 s) than the ejection time for DNA translocation from a single phage ($\leq$10 s) (*Grayson et al., 2007*) because ejection events at these conditions occur in a stochastic manner (*Freeman et al., 2016*) (if ejections were synchronized, the overall ejection time would be equivalent to DNA translocation time from a capsid). Thus, LS-measured scattering intensity decay reflects dynamics of ejection events versus time, discerning only between DNA-filled and empty capsids (*Freeman et al., 2016*). However, this observation does not exclude that a fraction of phages can eject DNA synchronously. As DNA ejection is progressing, each ejecting particle is given the same weight in the total scattering intensity signal change. Therefore, the signal from a phage population with fast synchronized ejection events occurring within seconds will be dominated by much slower desynchronized ejections occurring in parallel but on a much longer timescale of minutes. Something similar occurs in reaction kinetics measurements where resolution is determined by the slowest rate-limiting step. In our case, this will prevent differentiation between phage populations with synchronized and desynchronized ejection dynamics. Therefore, in order to be able to distinguish between synchronized and desynchronized ejection populations, a heavier weight must be assigned to the readout signal from the phage population with synchronized ejection dynamics. Such a measurement is possible with ITC-measured heat flow (i.e. rate of heat production measured in μJ/s or μcal/s or W) for the DNA ejection process when phage is titrated into LamB solution (*Liu et al., 2014*; *Jeembaeva et al., 2010*).

ITC is mainly used for thermodynamic analysis. However, it was recently demonstrated that it can be successfully used to measure reaction kinetics reflected by heat flow (so-called kinITC) due to the very short response time of modern ITC instruments (*Burnouf et al., 2012*). We designed a new assay using ultra-sensitive micro-ITC to determine ejection dynamics from phage λ with the aim to differentiate between populations with synchronized and desynchronized ejection events. We have previously shown that DNA ejection from phage λ is associated with an exothermic heat release, which can be accurately measured by ITC (*Liu et al., 2014*; *Jeembaeva et al., 2010*). Synchronized genome ejections will result in a significantly higher heat flow due to additive energy contributions from multiple ejecting λ-particles in one time instant, compared to that of desynchronized ejections where heat flow from single ejection events is spread over a longer period of time. Here, phage λsolution is titrated in LamB receptor solution with LamB present in a large excess ($10^4$:1 LamB: phage which ensures that ejection dynamics is not limited by receptor concentration). The instrument records differential power (DP) versus time (DP is equivalent to heat flow, measured in μcal/s) between the reference cell (containing buffer) and the sample cell (containing phage-LamB reaction mixture). *Figure 1* inset shows heat flow (μcal/s) versus time for the DNA ejection process occurring when wt-DNA length phage λ particles are titrated into LamB receptor solution at 22°C (all conditions remained identical to LS measurement for direct comparison). Heat flow contributions associated with enthalpy of mixing and pressure-volume work arising from titrating phage into buffer, buffer into LamB solution, and buffer into buffer are measured separately for each set of parameters and subtracted from the raw DP curve (*Liu et al., 2014*; *Jeembaeva et al., 2010*). All titration curves shown in this work (including *Figure 1* inset) are shown with background DP contributions subtracted.

The response time of our ITC instrument (Malvern iTC$_{200}$, see Materials and methods Section) is only ~3.5 s, which is significantly shorter than equilibration time for the DNA ejection process (e.g. it took ~250 s for the DP curve in *Figure 1* inset to return to baseline after titrating phage into LamB solution). This allows analysis of DNA ejection dynamics by directly evaluating the equilibration times of exothermic DP-peaks from phage-into-LamB titrations. (All separately-measured background DP contributions subtracted from the raw DP signal occur on a timescale of <10 s [*Liu et al., 2014*; *Jeembaeva et al., 2010*]) Here, we demonstrate how ITC provides information on population dynamics of synchronized and desynchronized ejection events not distinguishable by LS. As explained above, LS-measured scattering intensity in *Figure 1* reflects the total number of DNA-filled phage particles remaining over time as DNA ejections are progressing. On the contrary, ITC-measured heat flow in *Figure 1* inset corresponds to the instantaneous change in the heat released from a population of ejecting phage particles versus time. Thus, in order to compare the ejection dynamics from LS and ITC measurements, we need to take the derivative with respect to time of the LS-measured normalized forward scattering curve in *Figure 1*. This yields the instantaneous change in the number of ejecting phages versus time ($d\Delta I/dt$). The resulting $d\Delta I/dt(t)$ curve is scaled to

correspond to the phage concentration used in an ITC measurement and multiplied by the heat amount (enthalpy) released by one phage during ejection at a given temperature (*Liu et al., 2014*; *Li et al., 2015*). *Figure 1* inset shows the resulting LS scattering intensity derivative (blue curve) compared to the ITC titration curve (green curve). The derivate of LS data displays a slower decay than ITC data. As we have previously shown, LS overestimates the total ejection time since it also records the scattering intensity contributions from DNA condensates formed immediately after rapid DNA ejection (due to DNA ejection occurring faster than DNA diffusion in the bulk solution once DNA is ejected from the capsid) (*Freeman et al., 2016*; *Löf et al., 2007*). Ejected DNA condensates will relax in solution through diffusion, forming larger random coil aggregates with negligible contribution to the LS signal. This diffusive DNA coil relaxation process is masking the DNA ejection dynamics, making it appear slower (*Freeman et al., 2016*; *Löf et al., 2007*). The most marked difference, however, between LS derivative and ITC data in *Figure 1* inset is in the two exothermic peaks on the DP titration curve, which correspond to bimodal phage ejection dynamics from synchronized and desynchronized ejection events (in order to accommodate both slower LS data and ITC data on the time scale in *Figure 1* inset, the two ITC peaks are not clearly visible but they are well resolved when plotted on a shorter time scale as shown in the next section). This is not observed in the LS derivate displaying only averaged unimodal ejection dynamics. Thus, unlike LS scattering intensity, ITC heat flow data does not obscure the contribution from rapidly occurring synchronized ejections with slower decay from desynchronized ejections. This makes ITC a suitable method to investigate the effect of solid-to-fluid-like intra-capsid DNA transition on phage ejection population dynamics.

## Solid-to-fluid like DNA transition in the capsid controls phage ejection dynamics

As described above, we have recently found, using ITC (and supported by other methods), that DNA packaged in phage λ capsid undergoes a mechano-structural transition as a function of temperature (*Liu et al., 2014*), as illustrated in *Figure 2A*. By integrating the area under the exothermic DP titration curve for phage-in-LamB titration (with all mixing enthalpies and pressure-volume work subtracted), the enthalpy change associated with DNA ejection from phage can be derived and corresponds to $\Delta H_{ej}(T) = \Delta H(T)_{DNA\ ejected} - \Delta H(T)_{DNA\ inside\ phage}$. Since the total volume of the system does not change during DNA ejection, and the pressure is constant, enthalpy and internal energy are approximately equal (*Liu et al., 2014*; *Jeembaeva et al., 2010*). *Figure 2B* shows ITC-measured enthalpy $\Delta H_{ej}(T)$ for DNA ejection from wt phage λ when it is titrated into a solution with LamB receptor in the temperature range between 22°C and 42°C in 10 mM MgCl$_2$ Tris-buffer. *Figure 2B* shows a discontinuity in the approximately linear dependence of $\Delta H_{ej}$ on temperature occurring at $T^* \sim 33°C$ for wt λ-DNA length ejection from phage λ. The discontinuity is attributed to DNA structure disordering inside the capsid, which leads to a transition from a solid-like to a fluid-like state (*Liu et al., 2014*). The slopes in the linear regions of $\Delta H_{ej}(T)$ in *Figure 2B*, correspond to specific heat capacity $\Delta Cp(T)$ for DNA ejection process [$\Delta Cp(T) = Cp(ejected\ DNA) - Cp(packaged\ DNA)$]. Since change in $Cp(ejected\ DNA)$ (i.e. free DNA in solution) is relatively small in the measured temperature interval (22° - 43°C) (*Tsereteli et al., 2000*), the inversion of $\Delta H_{ej}(T)$ slope at transition temperature ($T^* \sim 33°C$) suggests a change in $Cp(packaged\ DNA)$ value for DNA inside the capsid. This $Cp$ change is in turn associated with transition in ordering and hydration of the packaged genome, which we have previously analyzed with ITC in ref. (*Jeembaeva et al., 2010*).

To investigate the effect of the DNA transition in λ-capsid on dynamics of the DNA ejection process, we analyze the equilibration time and median time of exothermic DP peaks on titration curves. *Figure 3A* shows ITC titration curves (shown as heat flow versus time) recorded for DNA ejection from wt phage λ at 22°C and 32°C (below the DNA transition temperature), and at 37°C and 42°C (above the DNA transition temperature) in 10 mM MgCl$_2$ Tris-buffer. *Figure 3A* shows that, above the DNA transition temperature (~33°C), only one enthalpy peak is observed, starting immediately after titration of phage in LamB (at time zero). However, the DP curve is not symmetrical and shows a right shoulder suggesting an unresolved second peak. Indeed, when the temperature was decreased below the transition temperature, the DP curve splits into two modes corresponding to two phage populations with different heat flow dynamics. At both 22° and 32°C, the first peak returns to baseline after ~20 s when phage is titrated into LamB, corresponding to a phage population with fast ejection dynamics. The second DP peak lasts for ~250 s at 22°C before returning to

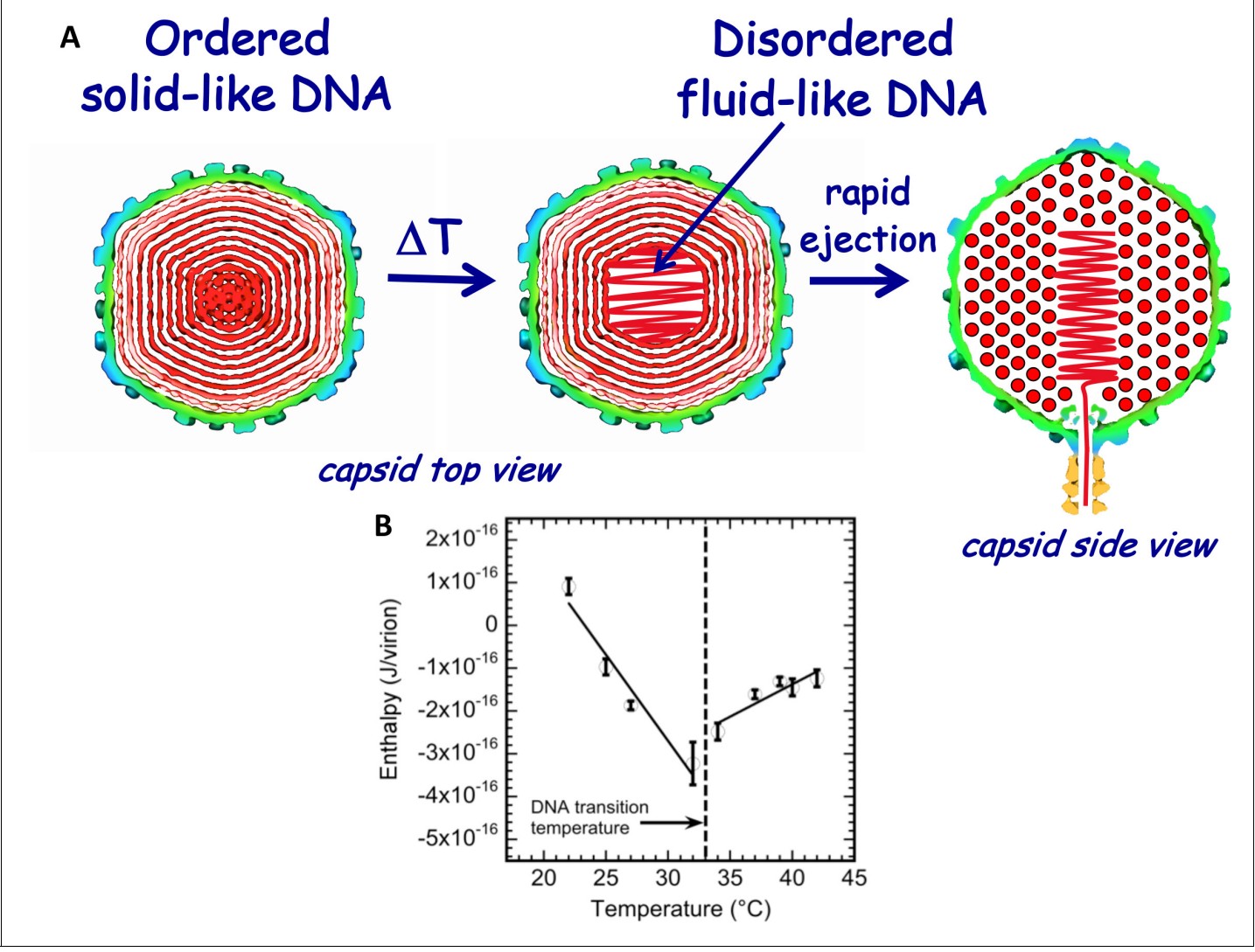

**Figure 2.** Schematic illustration of DNA disordering transition in the capsid. Transition occurs when the temperature is increased as we first observed with isothermal titration calorimetry. (**A**) DNA disordering transition, which occurs when the temperature is increased, originates close to the center of the capsid, where DNA is more stressed due to stronger bending and larger packing defects. (Left) Cross-sections of the top view of the capsid. (Right) A side view cross-section of the capsid, which illustrates that DNA closer to the center of the capsid is likely to be ejected first since, due to dsDNA bending stress constraints, it is the last DNA portion to be packaged in the capsid during phage assembly. The schematic illustration of DNA inside the capsid shows the ordering of an averaged DNA structure and not the arrangement of individual DNA strands. (**B**) Enthalpy of DNA ejection per virion (J) versus temperature for wt-DNA phage λ titrated into LamB solution. Abrupt enthalpy change inversion with temperature at ~33°C shows transition in energetics of DNA state inside the capsid. Dashed lines are drawn to guide the eye. $\Delta H_{ej}$ values were obtained as an average of six independent measurements for each sample. Vertical error bars are SEs.

DOI: https://doi.org/10.7554/eLife.37345.004

baseline value, corresponding to a phage population with slow ejection dynamics (the blue titration curve in *Figure 3A* is truncated at 150 s for clarity of presentation, however full ITC titration curve at 22°C returning to baseline after ~250 is shown in the *Figure 1* inset). Since these two exothermic peaks on a DP curve are not completely resolved, we have deconvoluted (*Goodman and Brenna, 1994*) the DP curves into two asymmetric Gaussian functions with an exponential damping term, using a precise deconvolution method (*Goodman and Brenna, 1994*) (see details in Supplemental Material Section). *Figure 3B* shows fitting of two exothermic DP peaks (dashed curves) for the DNA ejection process at 32°C. The same deconvolution fitting procedure was applied to the single asymmetric exothermic peaks with right skewed shoulder on the DP curves above the DNA transition

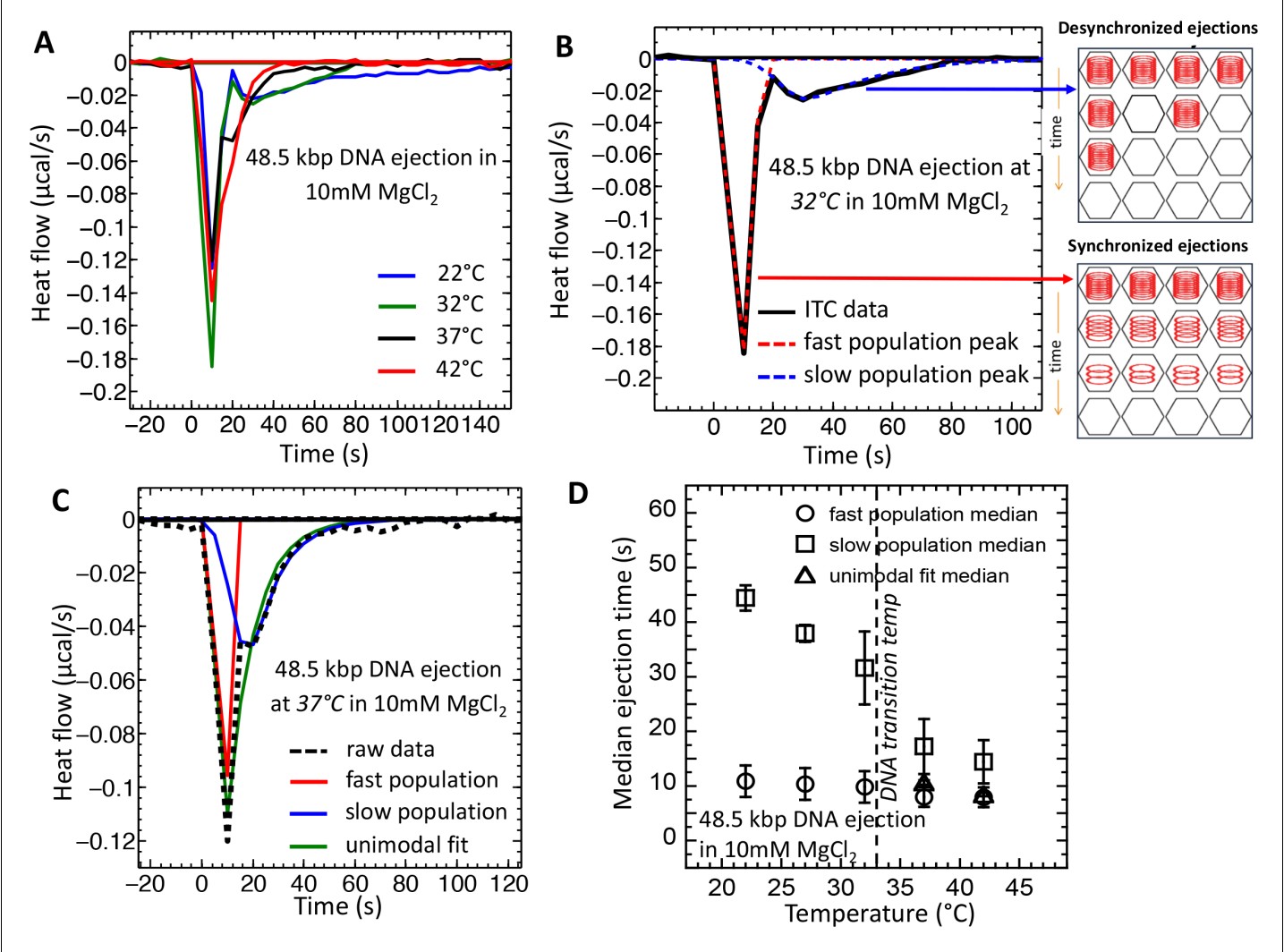

**Figure 3.** Temperature induced transition from two population (synchronized and desynchronized) to one population (synchronized) phage DNA ejection dynamics measured with ITC. ITC-measured heat flow (μcal/s) released when wt phage λis titrated into LamB solution, triggering DNA ejection at different temperatures, in 10 mM $MgCl_2$ Tris-buffer. (**A**) At temperatures below the intracapsid DNA transition (22 and 32°C), DP curve shows two exothermic peaks associated with two phage populations with fast (synchronized) and slow (desynchronized) DNA ejection dynamics. At temperatures above the intracapsid DNA transition (37 and 42°C), DP curve shows only one exothermic peak associated with one phage population with fast synchronized ejection dynamics. (**B**) DP curve for phage DNA ejection at 32°C (below DNA transition temperature) is fitted with deconvoluted skewed Gaussian curves, resolving peaks for both fast and slow ejection dynamics. (**C**) DP curve for phage DNA ejection at 37°C (above DNA transition temperature) fitted with skewed bimodal (deconvoluted) and unimodal Gaussian curves. Bimodal fit overlaps with unimodal fit, demonstrating that asymmetric unimodal Gaussian fit can be used to determine peak position and median time for ejection process above the DNA transition temperature, where one phage population with synchronized ejection dynamics is predominant. (**D**) Median time values derived from skewed Gaussian fit analysis for the DNA ejection process at five different temperatures (both bimodal and unimodal fits were made depending on the number of peaks). Error bars show the standard error (SE) of the width of Gaussian fit. Below the DNA transition temperature (22 and 32°C), two exothermic peaks are well resolved. Above the DNA transition temperature (37 and 42°C), deconvoluted peaks overlap and can be fitted with unimodal skewed Gaussian fit, as shown in (B).

DOI: https://doi.org/10.7554/eLife.37345.005

The following figure supplement is available for figure 3:

**Figure supplement 1.** Energy of activation for synchronized and desynchronized ejections measured by time-resolved SAXS.

DOI: https://doi.org/10.7554/eLife.37345.006

temperature in order to make a consistent comparison of DNA ejection dynamics at all measured temperatures. However, above the DNA transition temperature, two deconvoluted peaks derived from the fitting essentially converge into a single Gaussian, suggesting that there is mainly one dominant population of phage λ particles with faster ejection dynamics. This is illustrated in *Figure 3C*, which shows a DP curve for DNA ejection process at 37°C with deconvoluted bimodal and unimodal Gaussian fittings. Since most fitted Gaussian peaks on the DP curve are right skewed, we use the median value (instead of mode value) to estimate the central tendency time corresponding to average time for the overall DNA ejection process from a phage population. (In Supplemental Material Section we explain why the DP exothermic peaks are right skewed for DNA ejection process). *Figure 3D* summarizes median time values obtained from the Gaussian fits for DNA ejection from phage populations with fast and slow ejection dynamics at temperatures 22–42°C. The error bars are derived from the width of the Gaussian fits at the median value. *Figure 3D* shows that a phage population with fast ejection dynamics (corresponding to the first exothermic peak and referred to as 'fast population' in the plots) has an average ejection time of ~10 s at all temperatures. This ejection time corresponds to the time of DNA translocation from a single phage λ capsid under similar buffer conditions previously measured by single particle fluorescence (*Grayson et al., 2007*). This observation suggests that phage particles in the fast population eject their genomes immediately after receptor addition in a synchronized manner, which results in high heat flow (additive from many phage particles) and a sharp exothermic peak. On the other hand, the phage population with slow ejection dynamics (corresponding to the second exothermic peak and referred to as 'slow population' in all plots) is present only below the DNA transition temperature, where the slow ejection population coexists with the fast ejection population. Below the transition temperature, the average ejection time for the slow population is ~30 s at 22°C and ~45 s at 32°C (while DP peak equilibration to the baseline takes ~100 and ~250 s, respectively); see *Figure 3D*. These ejection times are significantly longer than DNA translocation time from a single phage (~10 s) (*Grayson et al., 2007*), suggesting that ejection events occur with delays in a desynchronized manner, as we had previously observed with LS (*Freeman et al., 2016*).

These observations demonstrate that a temperature-induced DNA transition from a solid- to a fluid-like state in λ capsid strongly affects the dynamics of genome ejection from phage. Below ~33°C, the temperature of intracapsid DNA transition, two exothermic peaks on the DP curve indicate ejection dynamics corresponding to two virus populations with synchronized and desynchronized ejections. The presence of synchronized and desynchronized populations suggests the co-existence of phage particles with packaged DNA in both fluid-like and solid-like states. This is supported by the fact that DNA is trapped in a metastable state while it is being packaged into the capsid due to high DNA-DNA electrostatic sliding friction, which leads to different non-equilibrium DNA conformations in the capsids (*Berndsen et al., 2014*; *Keller et al., 2016*). When temperature is increased above that of DNA transition (at $T > 33°C$), most of the phage population with slow desynchronized ejection dynamics is converted into a population with fast synchronized dynamics; the DP curve displays only one exothermic peak. The fast population corresponds to phage with DNA in a fluid-like state, where DNA-DNA electrostatic sliding friction is low (*Liu et al., 2014*) and genome ejection occurs immediately after phage is titrated into LamB solution.

In many dsDNA phages, there is a portal 'plug' protein at the portal-tail vertex that seals the portal opening through which DNA is ejected, facilitating retention of the packaged genome. For example, in phage P22, a homotrimer of the tail needle protein (gp26) acts as a portal plug (*Bauer et al., 2015*). Similarly in phage λ, protein gpW (the gp26 homolog) has been suggested to have a similar portal sealing function (*Bauer et al., 2015*; *Gaussier et al., 2006*). Adsorption of phage particles to cell receptors (LamB for phage λ) leads to portal plug opening, resulting in DNA ejection. As we have shown above, when DNA in the capsid is in a fluid-like pressurized state, phage-to-receptor adsorption results in immediate synchronized DNA ejections from all virions. By using time-resolved SAXS, we estimate the energy of activation for synchronized and desynchronized ejection processes by analyzing Arrhenius' dependence of DNA ejection rates on temperature (see data and method description in *Figure 3—figure supplement 1* and Materials and methods Section). The ejection rates are determined by measuring encapsidated DNA diffraction peak area versus time when phage λ is instantly mixed with LamB in a stopped-flow SAXS chamber. This reflects the number of remaining DNA-filled phage particles over time. We found that the energy of activation required to initiate the synchronized DNA ejection process from phage population with DNA in a fluid-like state is small,

with ejection rates essentially not influenced by temperature (*Figure 3—figure supplement 1*). On the contrary, the rates of DNA ejection events from phage with intracapsid DNA in a solid-like state show a strong dependence on temperature with relatively high activation energy (~20 times higher than molecular thermal energy) required to initiate ejection by overcoming a significant DNA-DNA sliding friction (*Berndsen et al., 2014*; *Liu et al., 2014*). This leads to stochastic delays in ejection events or even complete arrest of the ejection process at lower temperatures, as has been previously observed in vivo (*Mackay and Bode, 1976*) (e.g. no ejection occurs when phage λ is preadsorbed to *E. coli* cells at 4˚C). These observations suggest that measured activation energy for phage DNA ejection is associated with interstrand DNA sliding friction and mechano-strucutral transition of encapsidated genome, which further supports the ITC data above.

As we previously found (*Li et al., 2015*), the temperature at which the solid-to-fluid like DNA transition in phage λ occurs is strongly influenced by ionic conditions, since small polyvalent ions freely diffuse through the capsid pores and directly affect the DNA-DNA sliding friction and genome stress. Specifically, we had shown (*Li et al., 2015*) that, at physiologic temperature for phage infection (37˚C), varying ionic conditions will affect mobility of packaged DNA. As shown above, the transition from solid- and fluid-like DNA state in phage capsids results in desynchronized or synchronized DNA ejection dynamics, which in turn has been proposed to affect the cell decision between lysogenic and lytic infection (*Zeng et al., 2010*; *Trinh et al., 2017*). This raises an important biological question of how the extra- and intra-cellular ionic environment of a bacterial host affects ejection dynamics and consequentially the viral replication pathway. Therefore, in the next Section we investigate how deviations from ionic conditions mimicking those for optimum phage infectivity influence DNA ejection dynamics by switching between desynchronized and synchronized ejection behavior.

## Ion-regulated switch between synchronized and desynchronized ejection dynamics

Polyvalent cations present in the bacterial cytoplasm, such as polyamines and $Mg^{2+}$, have been shown to have a strong effect on the repulsive interactions between packaged DNA helices in a viral capsid through screening of negative DNA charges and/or introduction of attractive interaction (*Li et al., 2015*; *Evilevitch et al., 2008*; *Qiu et al., 2011*). Since free polyamine concentration in cells is very low as most polyamines are bound to cellular DNA and RNA (*Davis et al., 1992*; *Gibson and Roizman, 1971*; *Igarashi and Kashiwagi, 2000*), free Mg-ions will have a large effect on DNA interstrand interactions inside the capsid. Therefore, we are specifically interested in the effect of Mg-ion concentration on the dynamics of DNA ejection from phage λ at concentrations similar to those of extra- and intracellular $Mg^{2+}$ in vivo.

Mg-ions are essential for both cellular metabolism (e.g. enzyme activity, protein synthesis, preservation of ribosome and nucleic acid structures) (*Coates, 2010*; *Rodgers, 1964*; *Morgan et al., 1966*; *Lusk et al., 1968*) and the phage λ infectious cycle (*Fry, 1959*; *Gaussier et al., 2006*). It has been shown that a concentration of ~5–10 mM of $Mg^{2+}$ ions in the extracellular solution is critically important for both optimum adsorption of phage λ (*Fry, 1959*) to *E. coli*, as well as phage replication rate (*Mackay and Bode, 1976*; *Harrison and Bode, 1975*). A similar Mg-concentration has also been shown to provide optimum conditions for DNA packaging in phage λ (*Gaussier et al., 2006*; *Nurmemmedov et al., 2012*) and corresponds to the free Mg-concentration in *E. coli* cytoplasm (*Lusk et al., 1968*; *Kung et al., 1976*; *Hurwitz and Rosano, 1967*). As mentioned above, Mg-concentration affects DNA stress in the capsid and therefore strongly influences the temperature of transition between solid-like and fluid-like states (*Li et al., 2015*). At optimum temperature for infection (37˚C), this Mg-concentration should affect phage ejection dynamics leading to synchronized or desynchronized ejections depending on the intra-capsid DNA state. A correlation between physiologic Mg-concentration and phage λ genome ejection dynamics would suggest that mechanical intracapsid DNA transition is an important regulatory mechanism for viral replication. This will be discussed in Conclusions Section.

In the first set of measurements using ITC assay (*Figure 4A*), we determined DNA transition temperatures for wt λ-DNA length phage in $MgCl_2$ Tris-buffer with $MgCl_2$ concentrations varied between 5 and 50 mM. *Figure 4A* shows enthalpy $\Delta H_{ej}(T)$ versus temperature for the DNA ejection process triggered by phage λ titration into LamB solution. With increasing temperature, an abrupt

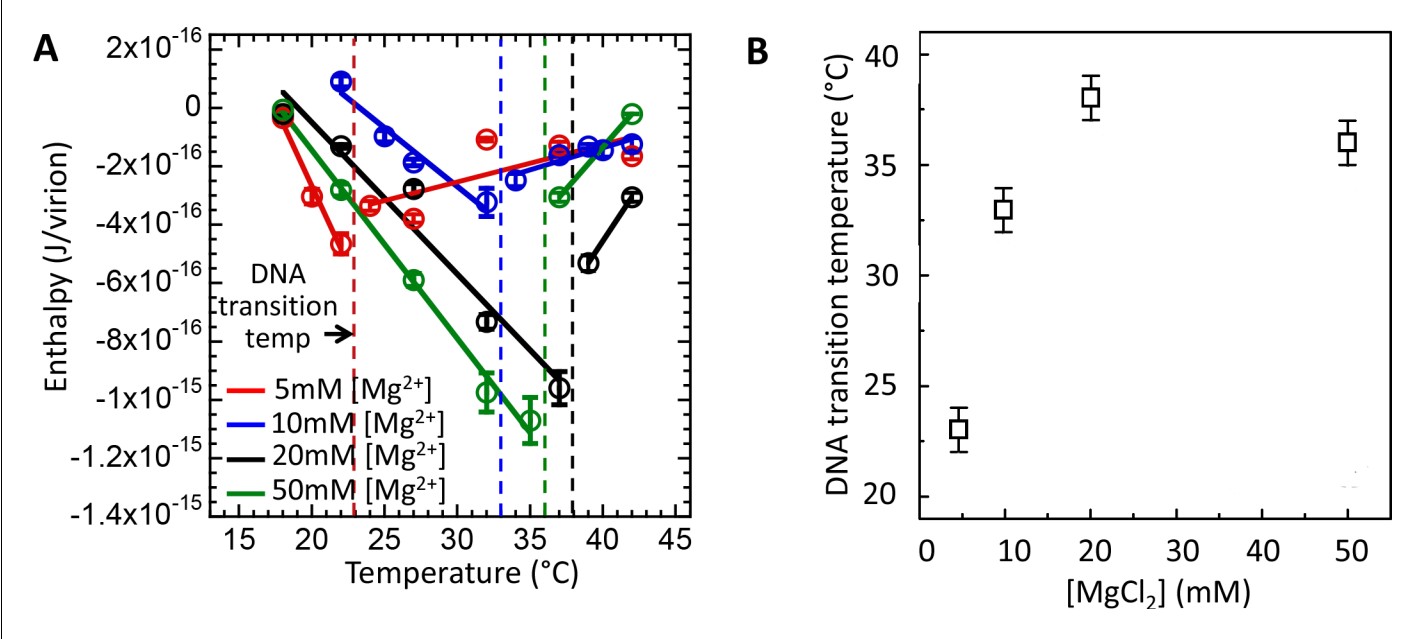

**Figure 4.** Intracapsid solid-to-fluid like DNA transition temperature as a function of Mg-concentration measured with ITC. Transition temperature $T*$ strongly rises with the initial increase in Mg-concentration but saturates at physiologic temperature of infection (37°C) at $[Mg^{2+}] \geq 20$ mM. (A) Enthalpy of DNA ejection per virion (J) from wt DNA phage λ versus temperature, $H_{ej}$, in $MgCl_2$ Tris-buffers with varying $MgCl_2$ concentrations (5 to 50 mM). Dashed lines are drawn to guide the eye. The vertical dashed lines indicate the DNA transition point. Vertical error bars are SEs. (B) Comparison between DNA transition temperatures determined by ITC as a function of $MgCl_2$ concentration in Tris-buffer.
DOI: https://doi.org/10.7554/eLife.37345.007

discontinuity and inversion in an essentially linear enthalpy change $\Delta H_{ej}(T)$ indicates the temperature $T*$ at which intracapsid DNA transition occurs. Increasing Mg-concentration will initially significantly reduce the strength of the interstrand repulsive interactions in the capsid due to counterion screening of the negative charges between packaged DNA helices (*Lander et al., 2013*; *Evilevitch et al., 2008*). However, we have previously shown that the screening effect of Mg-ions will become progressively smaller due to the counter-ion saturation (*Evilevitch et al., 2008*). As described above, DNA packaged in the capsid has to reach the critical stress level which increases with increasing temperature until the structural transition relieving the genome stress occurs at $T*$. Thus, reducing the intra-capsid DNA stress by increasing Mg-ion concentration leads to an increase in $T*$. That is, a higher temperature is required to reach the critical DNA stress limit in order to overcome the energetic barrier triggering the structural genome transition. Indeed, *Figure 4A and B* shows that the DNA transition temperature $T*$ strongly rises with the initial increase in Mg-concentration, from $T* \sim 23°C$ at 5 mM $MgCl_2$ Tris-buffer to $T* \sim 37°C$ at 20 mM $MgCl_2$ Tris-buffer. However, between 20 and 50 mM $MgCl_2$, $T*$ remains essentially unchanged due to Mg-ion saturation at the DNA helices (*Evilevitch et al., 2008*). As we have previously shown, at 5 mM $MgCl_2$ at 37°C (well above the DNA transition temperature), DNA in the λcapsid is in a fluid-like state (*Li et al., 2015*). However, at $[Mg^{2+}] \geq 10$ mM at 37°C, where the solid-to-fluid like DNA transition occurs at or close to this temperature, there is likely a coexistence between phage populations with DNA in fluid- and solid-like states inside the capsid.

Therefore, using ITC dynamics measurements, we investigate how a Mg-ion regulated switch between solid-like and fluid-like DNA states in λ-capsid affects DNA ejection dynamics at physiologic temperature of infection (37°C). Heat flow (μcal/s) versus time of exothermic peaks on DP titration curves was analyzed for phage λ into LamB titrations at 37°C and $Mg^{2+}$ concentrations between 5 and 50 mM. For clarity, *Figure 5A* shows selectively DP titration curves at 5 mM $Mg^{2+}$ Tris-buffer (above DNA transition) and 20 mM $Mg^{2+}$ Tris-buffer (below DNA transition). At 5 $MgCl_2$, where packaged DNA is in a predominantly fluid-like state (since $T* \sim 23°C$), only one exothermic DP peak corresponding to fast ejection dynamics is observed on the titration curve, with median ejection

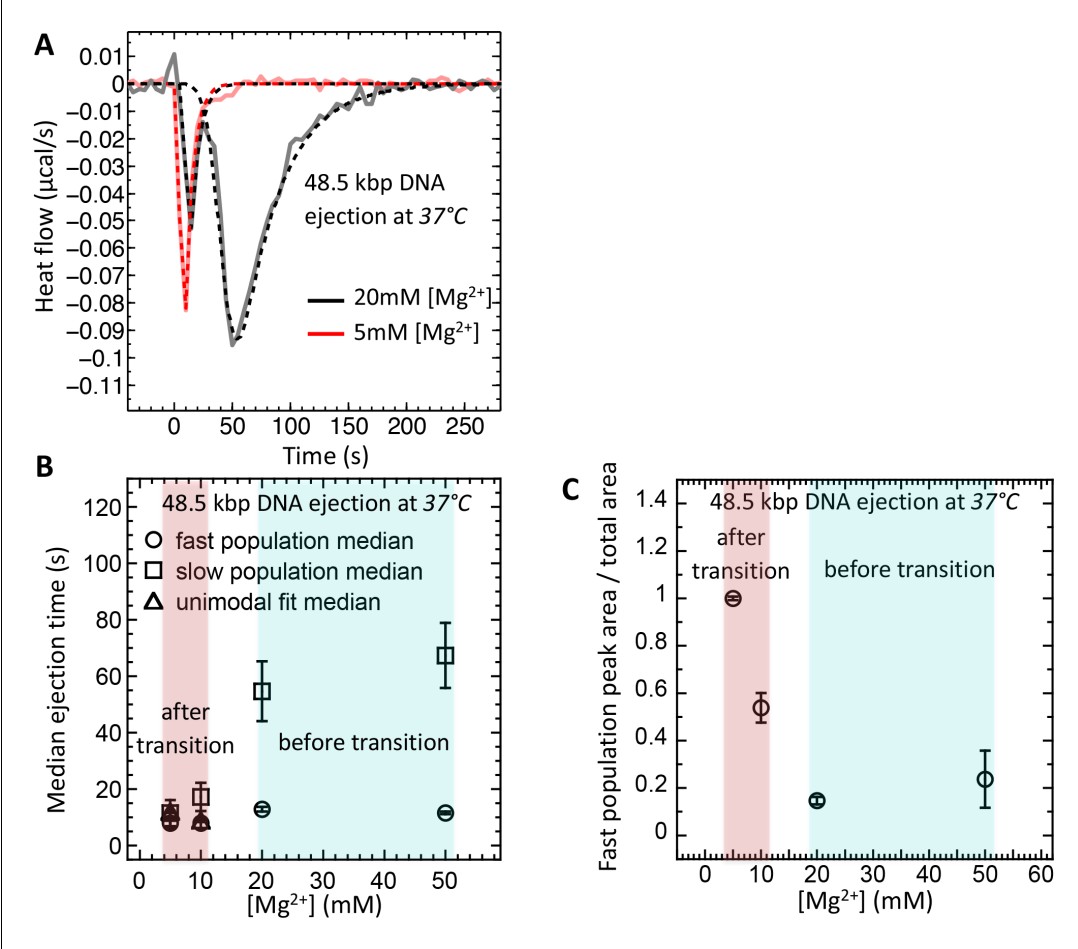

**Figure 5.** Well correlated with intracapsid DNA transition temperature, DNA ejection dynamics at physiologic temperature of infection (37˚C) is fast with synchronized ejection events at low [Mg²⁺] of < 20 mM (where packaged DNA is in a fluid-like state). At [Mg²⁺] ≥ 20 mM, there is a coexistence between phage populations with fast synchronized and slow desynchronized ejection dynamics (DNA in fluid- and solid-like states inside the capsid). Figure shows ITC measured heat flow (μcal/s) versus time for DNA ejection occurring at titration of phage λ into LamB solution at 37˚C and Mg²⁺ concentrations between 5 and 50 mM in Tris-buffer. (**A**) DP titration curves at 5 mM Mg²⁺ Tris-buffer (above DNA transition where intracapsid DNA is in a fluid-like state at 37˚C) and 20 mM Mg²⁺ Tris-buffer (below DNA transition where intracapsid DNA is in a solid-like state at 37˚C). At 5 MgCl₂, where packaged DNA is in a predominantly fluid-like state (since $T^* \sim 23$˚C), only one exothermic DP peak corresponding to fast synchronized ejection dynamics is observed on the titration curve, with median ejection time of ~10 s (corresponding to DNA translocation time from a capsid). At 20 mM MgCl₂, the intracapsid DNA is at its transition temperature ($T^* \sim 37$˚C) and the DP titration curve shows two distinct exothermic peaks attributed to fast and slow ejection dynamics corresponding to coexisting phage populations with intracapsid DNA in solid-like and fluid-like states. (**B**) Median ejection times (corresponding to average ejection times) derived from deconvoluted skewed Gaussian fits to DNA ejection DP peaks in 5–50 mM MgCl₂ Tris-buffers at 37˚C. (**C**) The ratio between fast population peak area (for synchronized ejection events) and total area of both fast and slow exothermic peaks. This plot reflects the fraction of phages (from the whole phage population) with intracapsid DNA in a fluid-like state and synchronized ejection dynamics. Figure shows how variation in Mg-concentration has a dramatic effect on ejection dynamics by switching it from all synchronized to nearly all desynchronized DNA ejection events.

DOI: https://doi.org/10.7554/eLife.37345.008

time of ~10 s, see *Figure 5A*. As discussed above, this average ejection time corresponds to DNA translocation time from a single phage capsid and reflects synchronized ejection events from a majority of phage population. On the contrary, at 20 mM MgCl₂, the intracapsid DNA is at its transition temperature ($T^* \sim 37$˚C). Now, the DP titration curve shows two distinct exothermic peaks attributed to fast and slow ejection dynamics. The first peak has median ejection time of ~10 s corresponding to a phage population with DNA in a fluid-like state and synchronized ejection dynamics. The second peak (corresponding to a phage population with DNA in a solid-like state) has median ejection time of ~55 s suggesting stochastic delays in initiation of DNA ejection resulting in

slower ejection dynamics. In *Figure 5B*, we summarize these data by plotting median ejection times (corresponding to average ejection times) derived from deconvoluted skewed Gaussian fits to DNA ejection DP peaks in 5–50 mM $MgCl_2$ Tris-buffers at 37°C. To further quantify these results, in *Figure 5C* we show the ratio between fast population peak area (for synchronized ejection events) and total area of both fast and slow exothermic peaks. Assuming that ejection enthalpy per virion is approximately the same for phage with DNA in a fluid- or in a solid-like state, this plot reflects the fraction of phages (from the whole phage population) with DNA in a fluid-like state and synchronized ejection dynamics. The observed fraction of fast phage population with synchronized ejections varies between 100% (at 5 mM $Mg^{2+}$) and only ~10% (at $[Mg^{2+}] \geq 20$ mM). Thus, *Figure 5C* shows how variation in Mg-concentration at 37°C has a dramatic effect on ejection dynamics from phage λ by switching from all synchronized to nearly all desynchronized ejection events. Consequentially, at Mg-concentrations above that of physiologic optimum for infection (~5–10 mM), DNA ejection dynamics is significantly reduced with an order of magnitude slower average ejection time compared to that of a fast population with synchronized ejections.

## Conclusions

Different bacteriophage traits determine the dynamics and heterogeneity of viral populations. Investigating these traits is of fundamental importance for understanding viral replication pathways and for improving the potency of treatments and viral vaccines (*Lee et al., 1997*), and studying viral fitness (*Goldhill and Turner, 2014*). Structural virology has been instrumental in providing a detailed morphological description and classification of viruses and their receptors required for studies of traits influencing virus-host interaction dynamics (*Gallet et al., 2011*). However, most of the structural studies have been focused on viral capsids due to their symmetry, rather than on the encapsidated genome where the lack of symmetry prevents high-resolution analysis. Recent findings of pressure-driven viral DNA ejection into cells from phage (*Evilevitch et al., 2003*; *Castelnovo and Evilevitch, 2007*), Archaeal viruses (*Hanhijärvi et al., 2013*) and Herpesviruses (*Bauer et al., 2013*), have raised strong interest in understanding the role that packaged dsDNA structure plays in dynamics of genome ejection and viral replication. However, the structure of the encapsidated genome and its mechanical properties have until now not been sufficiently investigated nor considered as a trait influencing viral population dynamics. While this work is an in vitro study, it suggests a novel mechanism affecting viral replication dynamics that can influence the cell's decision between lytic and lysogenic infection, prompting further investigations in vivo.

In this work, we discovered, for the first time, a direct link between the structure (and its associated mobility) of the encapsidated dsDNA in phage λ and dynamics of initiation of its release from the capsid. We demonstrate that the solid-to-fluid-like intracapsid DNA transition (*Liu et al., 2014*), is a determining factor for either rapid synchronized or slow desynchronized ejection dynamics from a phage population. We show that at optimum temperature for infection (i.e. ~37°C) and Mg-concentration $\leq$ 10 mM, the encapsidated DNA is in a fluid-like state and ejection events from the majority of the phage population are synchronized with DNA ejections occurring within ~10 s directly after phage adsorption to *E. coli* receptors (this time corresponds to DNA translocation time from a capsid). However, deviations from these conditions for infection (e.g. low temperature, high Mg-concentration) result in intracapsid DNA being in a solid-like state where DNA is arrested in different conformations due to high interstrand sliding friction (*Berndsen et al., 2014*; *Liu et al., 2014*). This leads to a striking heterogeneity in phage ejection dynamics, displaying coexistence of populations with synchronized ejection events (where DNA is in a fluid-like state) and desynchronized stochastically occurring ejection events (where DNA is in a solid-like state). The latter presents one to two orders-of-magnitude slower ejection dynamics, ranging from minutes to tens of minutes, depending on salt conditions and temperature, compared to DNA translocation time from a capsid. This ejection dynamics behavior is in good agreement with earlier observations of temperature's effect on DNA injection dynamics from phage λ after preadsorption to *E. coli* culture (*Mackay and Bode, 1976*). Our findings also dismiss an earlier claim that stochasticity of DNA ejections from phage is solely controlled by capsid portal opening dynamics (*Raspaud et al., 2007*; *Frutos et al., 2016*). These assumptions were based on earlier LS measurements of phage ejection dynamics that could not distinguish between synchronized and desynchronized ejection events, as we have demonstrated above. (The same reference [*Frutos et al., 2016*]) also incorrectly concludes that there is no

intracapsid DNA transition in phageλ with increasing temperature. As described above, we have clearly demonstrated that solid-to-fluid like DNA transition occurs as a function of salt and temperature, signified by an abrupt change in mobility (measured with AFM nanoindentation) and internal energy (measured with ITC) of the packaged λ-genome (*Li et al., 2015*; *Liu et al., 2014*).

As discussed above, it was recently proposed that the lysogeny-lytic cell decision initially occurs at the phage level through the timing of DNA delivery from multiple phages into a cell during infection (*Zeng et al., 2010*; *Trinh et al., 2017*). The timing of DNA injection from phage is what determines the onset of viral genome replication because both phage DNA injection and DNA replication processes occur on comparable timescales (*Mackay and Bode, 1976*). In turn, replication dynamics of injected phage genomes determines how phages interact with one another (*Trinh et al., 2017*) when multiple phage particles are infecting a single cell. It was shown that lysogeny is more likely to occur when infecting phages interact cooperatively during cell infection (*Kourilsky, 1973*). Synchronized injections result in immediate presence of DNA from multiple phages in a cell that can be integrated into cell's chromosome without competition for the progeny (*Zeng et al., 2010*; *Trinh et al., 2017*). Lysogeny results in the absence of selection, which allows various mutants to integrate and persist in the lysogenic state, leading to diversification of phage genes and adaptation to poor or new growth conditions (*Zeng et al., 2010*; *Trinh et al., 2017*). On the contrary, when DNA injections are desynchronized, few phage particles are delivering their genes faster than the rest of the phage population, resulting in competitive interactions. If first phage to deliver its genome is lytic, it will override lysogeny, rapidly replicate itself using the cell's limited resources and therefore take a larger share of phage progeny. As mentioned above, the asynchronous infection delays decrease the chance of lysogeny by also lowering CII viral transcription factor levels in the cell, where CII controls cell's decision (*Cortes et al., 2017*). It was shown that the cutoff time, where delayed phage infections of one cell no longer affect the lytic-lysogenic cell decision varies from few minutes to tens of minutes, depending on the MOI (higher MOI results in shorter cutoff time) (*Cortes et al., 2017*), comparable to timescales for overall ejection delays observed in our measurements. This suggests that ejection delays resulting from solid-like DNA state in the capsid will incease cell lysis probability.

The lytic-lysogenic decision is one of the most crucial factors determining virus population dynamics. In general, it was observed that limited conditions for phage replication favor lysogeny (*Zeng et al., 2010*; *Kourilsky, 1973*). This is explained by the fact that lysis is metabolically more 'expensive' and therefore would not be successful if the cells are not sufficiently healthy. Here, we found that ionic composition in the surrounding medium of bacterial cells acts as an on-off switch for DNA structural transition in λ-capsids leading to either synchronized or desynchronized ejection events at the optimum temperature for infection (37°C). It was previously shown that starvation of $Mg^{2+}$ stimulates lysogenization (*Kourilsky, 1973*). Our study shows that, at low Mg-concentration, intracapsid DNA stress is high, leading to a DNA transition from solid- to fluid-like at temperatures well below that of infection temperature (37°C). This DNA transition results in synchronized ejections at 37°C, which indeed stimulate lysogenization, as we have discussed immediately above. Remarkably, both of these in vitro and in vivo (*Kourilsky, 1973*) observations are in good agreement, suggesting that DNA injection dynamics from phage into cells may play a significant role in the cell fate decision. However, while the probability of lysogeny depends inversely on phage infection delays (*Cortes et al., 2017*), the infection timings alone cannot be used to predict the outcome of either lysogenic or lytic behavior since other factors (e.g. cell volume [*St-Pierre and Endy, 2008*]) have also been shown to play a role. Furthermore, we have limited understanding of how many host functions (e.g. HflKC, cAMP, etc) feed into the decision (*Casjens and Hendrix, 2015*). Mapping all host functions into a single axis of rich/poor conditions is not supported by experiments. Thus, further interpretation of the effect of desynchronized and synchronized infections on the cell decision process would require a new model for circuit dynamics.

In conclusion, these in vitro findings suggest a new paradigm affecting virus population dynamics—mechano-signaling mediated by packaged DNA structure in phage λ. The environmental factors regulating the mechanical transition between solid- and fluid-like intracapsid DNA states strongly influence the timing of genome ejections. In vivo, desynchronized ejections from a phage population lead to competition between phage genomes, affecting the rate of phage gene replication and transcription as well as the lysogeny-lytic cell decision. Given similarities between phageλ and Herpesviruses in their mechanism of pressure-driven DNA ejection (where a temperature-induced solid-to-

fluid like DNA transition was observed in both viral systems [*Sae-Ueng et al., 2014*; *Bauer et al., 2013*]), the new insights into the lysogeny-lytic decision process for phage provide intriguing prospects for understanding of latency mechanism in Herpesviruses. Indeed, analogies between phage and Herpesviruses in other regulatory factors that affect viral latency have been previously observed (*Pai and Weinberger, 2017*; *Weller and Sawitzke, 2014*).

## Materials and methods

### Bacteriophage λ and LamB receptor

Following previous protocols (*Evilevitch et al., 2003*), phage λ strain λ cl857 with a wild type (wt-) genome length of 48.5 kbp were produced by thermal induction of lysogenic Escherichia coli (*E. coli*) strain AE1 derived from S2773 strain. After cell harvest, phage particles were precipitated in 10% polyethylene-glycol (PEG) 8000 and purified by CsCl equilibrium centrifugation. Later on phages were dialyzed against TM-MgSO$_4$ buffer (10 mM MgSO$_4$, 50 mM Tris HCl, pH7.4) or TM-MgCl2 (10 mM MgCl$_2$, 50 mM Tris HCl, pH7.4) for overnight at 4°C. The receptor protein LamB was expressed and purified from pop-154, an *E. coli* K12 strain in which the LamB gene was transduced from Shigella sonnei 3070. The detailed preparation method was described earlier (*Ivanovska et al., 2007*).

### Isothermal titration calorimetry (ITC)

Isothermal titration calorimetry (ITC) was used to measure the ejection enthalpy of λ genome, $\Delta H_{ej}$. All ITC measurements in this study were performed using the MicroCal iTC200 system manufactured by Malvern. The principles of measuring DNA ejection enthalpy in phages λ were described before (*Jeembaeva et al., 2010*). In ITC experiment, 2.69 μL λ particles at $10^{12}$ - $10^{13}$ pfu/mL (10 – 100 nM) were titrated into 200 μL LamB solution in the sample cell (reference cell was always filled with MilliQ water). LamB is at a concentration of 0.2 - 0.5 mg/mL (1.4 - 3.5 μM). The molar ratio between LamB trimmers and λ particles in the sample cell is always kept above $10^4$:1 (up to 3 titrations) to ensure the maximum ejection efficiency of λ with no delay time. Both LamB and λ particles were in the same dilution buffer containing TM (10mM MgCl$_2$ or MgSO$_4$, as indicated, 50mM Tris, pH 7.4) and 1% oPOE. The dilution heat of phage particles, LamB particles were measured separately and excluded from the final value of $\Delta H_{ej}$ (*Jeembaeva et al., 2010*). The ejection enthalpy, $\Delta H_{ej}$ is a good estimate of the internal energy of λ genome, $U_{encap\_DNA}$. In an ITC experiment, $\Delta H_{ej}$ is measured as an enthalpy change associated with the genome release process, $\Delta H = \Delta U + p\Delta V$, where $\Delta U$ is the change in internal energy of the system during DNA ejection ($\Delta U = U_{empty\_cap} - U_{filled_{cap}} = U_{encap\_DNA}$), $p$ is the atmospheric pressure acting on the system and $\Delta V$ is the change in volume of the solution surrounding the phages during DNA release. Since this volume change in solution, $\Delta V$ is negligible, $\Delta H \approx \Delta U$.

### Phage λ DNA ejection dynamics measured by time-resolved SAXS

Time-resolved small angle x-ray scattering (SAXS) measurements were carried out at the 12-ID B station at the Advanced Photon Source (APS) at Argonne National Laboratory. A 12KeV x-ray beam was used to illuminate the sample with a short sample-to-detector distance (1 m) and an overall q range from 0.013 to 1.90 Å$^{-1}$. To monitor the DNA ejection process, we mixed 60 μL of phage solution with 120 μL of LamB solution in a three-way static mixing tee, the mixture was then sent to a flow-through glass capillary perpendicular to the incident x-ray. The mixing ratio between phage and LamB was kept as 1 to 200. To reduce photon damage, the sample solution was oscillating during the SAXS measurement with a flow rate of 10 μL/s. The SAXS pattern was recorded every 2 to 10 s with an x-ray exposure time of 1 s. After converting the 2D SAXS pattern to 1D Intensity versus q plot, the DNA peak between 0.18 to 0.32 Å$^{-1}$ was truncated and analyzed. Since LamB and 1% oPOE micelles also contribute to the overall scattering intensity, it is important to properly subtract their intensity before analyzing the DNA peak. The SAXS profiles for TM buffer, 1% oPOE in TM buffer, undiluted LamB in 1% oPOE TM buffer were collected at different temperatures. The background was constructed using a linear combination of TM buffer, 1% oPOE and LamB. After background subtraction, the DNA peak was fitted to a Gaussian plus a linear function. Detailes for DNA peak fitting procedure can be found in our previous publications (*Liu et al., 2014*).

## Energy of activation for synchronized and desynchronized ejections

Solution SAXS provides structural information that reveals base pair (bp) ordering and average DNA-DNA spacing of the encapsidated genome (*Earnshaw and Harrison, 1977*; *Liu et al., 2014*). However, time-resolved SAXS measurements of DNA structure in phage capsids during the LamB-triggered DNA ejection process can directly reflect the ejection dynamics. *Figure 3—figure supplement 1A* shows integrated scattering intensity, $I$, versus scattering vector $q$ for wt DNA phage λ particles. In the lower $q$ region (0.007 to 0.1 $\text{Å}^{-1}$), the scattering profile originates from the highly symmetrical icosahedral phage capsids. At higher $q$ (between 0.2 to 0.3 $\text{Å}^{-1}$), the single peak with the small oscillating ripples on its top is due to the diffraction from the encapsidated ordered DNA strands. The short-range DNA-DNA interaxial spacings determine the position of the DNA diffraction peak (*Earnshaw and Harrison, 1977*). The area of DNA peak indicates how well the DNA strands are aligned relative to each other since it is directly proportional to the total number of ordered DNA bp in the capsid (*Earnshaw and Harrison, 1977*; *Liu et al., 2014*). During DNA ejection from phage, DNA diffraction peak area is therefore progressively decreasing due to reduced bp ordering resulting from less coherent diffraction. (The position of the peak should then also gradually shift to lower $q$ values due to increasing distance between packaged DNA strands).

To investigate DNA ejection dynamics' dependence on temperature, we measured DNA diffraction peak area versus time when phage λ is instantly mixed with LamB in a stopped-flow SAXS chamber. This reflects the number of remaining DNA-filled phage particles over time. As discussed above, DNA ejection dynamics was previously evaluated by measuring total forward scattering intensity over time, $I(0)$, using LS (*Figure 2A* in the main text) (*Freeman et al., 2016*) since $I(0)$ depends on density of DNA packaged in the capsid. However, the measured decay in $I(0)$ versus time also depends on diffusive relaxation time of DNA coils that initially remain condensed in solution immediately after ejection (due to DNA ejection rate being faster than DNA diffusive relaxation in bulk solution) (*Freeman et al., 2016*; *Löf et al., 2007*). One of the advantages of following genome ejection dynamics by using SAXS to measure the decrease in DNA diffraction peak area (instead of measuring change in forward scattering intensity), is in the reduced signal contribution from this secondary process of slow diffusion-relaxation dynamics of ejected DNA coils. This is due to the fact that DNA diffraction peak area depends mainly on bp ordering with defined DNA-DNA spacing inside the capsid (bp ordering is mainly lost once DNA is ejected), as opposed to forward scattering intensity reflecting condensed DNA density (rather than ordering) both inside and outside the capsid. The difference in measured DNA ejection dynamics determined by LS (forward scattering $I(0)$), SAXS (forward scattering $I(0)$) and decay in SAXS-measured DNA diffraction area is illustrated in *Figure 1* in the main text (a comparison is also made for derivatives of these three types of data compared with ITC DP curve in *Figure 1* inset). *Figure 3—figure supplement 1B* shows time-resolved DNA diffraction peak for the DNA ejection process from wt DNA length phageλ in TM-buffer at 25° C (below the DNA transition temperature of ~33°C). When the majority of phage particles have ejected their genomes, the DNA diffraction peak disappears due to lack of ordered DNA. As we observed with ITC-measured dynamics above, the decay time for DNA diffraction peak area at 25°C is much longer than the time for DNA translocation from a single phage particle (~10 s), suggesting that DNA ejection events are occurring in both a synchronized and desynchronized manner. Thus, on the timescale of this measurement, the positon of DNA diffraction peak remains essentially unchanged with time since it mainly shows the decay in number of completely DNA-filled phage capsids.

In order to estimate the energy of activation for phage populations with synchronized and desynchronized ejections, we analyzed the Arrhenius' dependence of the rate of ejection events on temperature. The natural log of DNA diffraction peak area was plotted versus time in *Figure 3—figure supplement 1C* for the DNA ejection process at 25, 30, and 37°C. Interestingly, inset in *Figure 3—figure supplement 1C* shows that, during the first ~20 s, the ejection rate has almost no temperature dependence. However, at longer times, strong temperature dependence is observed (*Figure 3—figure supplement 1C*). As we found with ITC in *Figure 3* in the main text, the first 20 s of the DNA ejection process are dominated by synchronized ejection dynamics from a phage population with intracapsid DNA in a fluid-like state. The lack of SAXS-measured ejection rate dependence on temperature during the first 20 s suggests that ejection dynamics initially does not display a measurable activation energy barrier for DNA ejection once the capsid portal is opened through

receptor adsorption. Such a scenario is plausible, since packaged DNA is in a fluid-like state, where it has low interstrand friction leading to low activation energy required to initiate ejection. This also suggests that activation energy for portal plug opening in the capsid is relatively small, being strongly reduced when LamB binds to the phage tail, as we previously observed (*Freeman et al., 2016*; *Bauer et al., 2015*). However, at time > 20 s, strong temperature dependence is observed in SAXS-measured DNA ejection rate (*Figure 3—figure supplement 1C*), suggesting measurable energy of activation is required for initiation of DNA ejection from phage. In agreement with this observation, we found with ITC (*Figure 3* in the main text) that, at > 20 s, DNA ejection events are desynchronized below DNA transition temperature (~33°C), corresponding to the second exothermic DP peak originating from a phage population with DNA packaged in a solid-like state. These observations suggest that measured activation energy is mainly associated with DNA-DNA friction that must be overcome to initiate genome ejection from the capsid (*Gabashvili and Grosberg, 1992*; *Berndsen et al., 2014*).

To quantify energies of activation for a phage population with DNA in solid-like and fluid-like states, in *Figure 3—figure supplement 1D* we plotted natural logarithms of the ejection rate constants (*k*) (corresponding to the slopes of the linear fits in *Figure 3—figure supplement 1C*) as a function of inverse temperature. In order to separate the rates for synchronized and desynchronized ejection dynamics, the rate constants in *Figure 3—figure supplement 1D* are plotted separately for the fast population at time interval < 20 s, and for the slow population at times > 20 s. The ejection rates for the slow population ejection dynamics (> 20 s) show exponential rate dependence on inverse temperature (i.e. Arrhenius dependence), evidenced by the linearity of the fits in *Figure 3—figure supplement 1D*. Energy of activation derived from the linear fit for the slow population is ~$1.2 \times 10^{-19}$ J/virion corresponding to ~29 kT at 25°C. Thus, activation energy to initiate ejection from phage λ with intracapsid DNA in a solid-like state is relatively high (~20 times higher than molecular thermal energy) due to a significant DNA-DNA sliding friction during initiation of the ejection process when interstrand separation is small (*Berndsen et al., 2014*; *Liu et al., 2014*). This in turn leads to stochastic delays in ejection events or even complete arrest of the ejection process at lower temperatures, as has been previously observed in vivo (*Mackay and Bode, 1976*) (e.g. no ejection occurs when phage λ is preadsorbed to *E. coli* cells at 4°C). *Figure 3—figure supplement 1D* also shows that, for a fast population with synchronized ejection dynamics and intracapsid DNA in a fluid-like state (at times < 20 s), the ejection rate dependence on inverse of temperature shows essentially no or little temperature dependence. It can be noted here that slight temperature dependence of DNA ejection rate below 20 s can be attributed to activation energy associated with portal opening (*Freeman et al., 2016*). However, for a slow population with desynchronized ejection dynamics (where intracapsid DNA is in a solid-like state), this is not the dominant component of activation energy (*Bauer and Evilevitch, 2015*), since the main contribution comes from DNA-DNA sliding friction, as suggested by this data.

## Gaussian peak deconvolution

Some ITC titration curves (DP versus time) in our study exhibit bimodal population dynamics with not fully resolved exothermic titration peaks. We can mathematically deconvolute the experimental curve into two asymmetric Gaussian functions with an exponential damping function. These Gaussians were termed as Exponentially Modified Gaussian distribution (EMG) described by $\frac{a}{2b} \cdot \exp\left(\frac{c^2}{2b^2} + \frac{d-t}{b}\right) \cdot \left[\mathrm{erf}\left(\frac{t-d}{c\sqrt{2}} - \frac{c}{\sqrt{2}b}\right) + 1\right]$ (*Grushka, 1972*). This deconvolution method was previously described as high precision method (*Goodman and Brenna, 1994*). Equation parameter a denotes the peak area from the deconvoluted Gaussian peak, b denotes the exponential damping term (exponent relaxation time), with b = 0 corresponding to a symmetric Gaussian curve. c indicates the width of the Gaussian, which provides the error bars in our figures. d is the position of an individual Gaussian peak. The estimated population mean of EMG is shifted slightly from the peak position. It is the peak position plus the exponential relaxation time μ = d + b. The population standard derivation is $s^2 = c^2 + b^2$ (*Grushka, 1972*). Skewness of the distribution is defined as $\gamma = \frac{2b^3}{(c^2+b^2)^{3/2}}$. The median of the distribution equals to μ - 1/3sγ (according to Pearson's second skewness law).

## Shape of ITC titration peaks corresponding to DNA ejection process from phage

ITC measured exothermic titration peaks can be nicely fit with asymmetric Gaussian functions. As explained in the manuscript, the first exothermic DP peak results from synchronized DNA ejections for a phage population. The shape of this peak will depend on enthalpy change associated with simultaneous DNA translocation from multiple phage particles. It has been shown that DNA translocation speed increases with ejection time at the beginning due to a decrease in the DNA-DNA frictional force as DNA packing density decreases in the capsid (*Grayson et al., 2007*). However, the translocation speed starts to decrease towards the end of the ejection process due to decrease in the DNA pressure driving its ejection from the capsid. This variation in DNA translocation rate is reflected by the rate of internal energy dissipation (heat flow measured by ITC) during the ejection process which in turn is fitted by an asymmetric Gaussian function. The asymmetric Gaussian shape of the second exothermic DP peak on the titration curves, corresponding to stochastic DNA ejection events results likely from both distribution of delayed particle ejection modes and non-uniform rates of DNA translocation.

## Acknowledgements

Ting Liu is greatly acknowledged for help with data analysis. Dong Li and Xiaobing Zuo are acknowledged for help with supporting SAXS data collection for the Supplemental *Figure 3—figure supplement 1*. The author is grateful to Krista Freeman, Ido Golding, Lanying Zeng, Sherwood Casjens, Ian Molineux and Philippe Dumas for important feedback and discussions in the course of preparation of this manuscript. Funding for this work was provided by the National Science Foundation CHE-1744061 (to AE) and Swedish Research Council grants (Vetenskapsrådet) 621-2014-5537 and 349-2014-3962 (to AE).

## Additional information

### Funding

| Funder | Grant reference number | Author |
|---|---|---|
| National Science Foundation | CHE-1744061 | Alex Evilevitch |
| Vetenskapsrådet | 62120145537 | Alex Evilevitch |
| Vetenskapsrådet | 34920143962 | Alex Evilevitch |

Funding for this work was provided by the National Science Foundation CHE-1744061 (to AE) and Swedish Research Council (VR) grants 621-2014-5537 and 349-2014-3962 (to AE).

### Author contributions

Alex Evilevitch, Conceptualization, Resources, Data curation, Formal analysis, Supervision, Funding acquisition, Validation, Investigation, Visualization, Methodology, Writing—original draft, Project administration, Writing—review and editing

### Author ORCIDs

Alex Evilevitch https://orcid.org/0000-0002-0245-9574

### Decision letter and Author response

Decision letter https://doi.org/10.7554/eLife.37345.011
Author response https://doi.org/10.7554/eLife.37345.012

## Additional files

### Supplementary files

• Transparent reporting form

DOI: https://doi.org/10.7554/eLife.37345.009

**Data availability**

All data generated or analysed during this study are included in the manuscript and supporting files.

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
