## [Decision Letter]

Thank you for submitting your article "Do phages infect in unison? The mobility of packaged phage genome controls ejection dynamics" for consideration by *eLife*. Your article has been reviewed by three peer reviewers, and the evaluation has been overseen by a Reviewing Editor and Arup Chakraborty as the Senior Editor. The following individual involved in review of your submission has agreed to reveal his identity: Michael Feiss (Reviewer #2).

The reviewers have discussed the reviews with one another and the Reviewing Editor has drafted this decision to help you prepare a revised submission.

Summary:

A great deal of work has been done, over several decades, on the issue of what determines the lytic versus lysogenic pathway of an infecting bacterial virus. Most recently, experiments have correlated asynchrony versus synchrony of infection by multiple phages with lysis versus lysogeny, suggesting that phages compete for replication resources in the former case and cooperate in bacterial-chromosome-integrated coexistence in the latter case. Earlier experiments had suggested that the time scale associated with genome ejection/initiation of infection, following the binding of phages, is controlled predominantly by temperature (but also by ionic strength), being essentially immediate at, say, 37°C (for λ and *E. coli*), but involving time lags of 5-10 minutes at 25°C, and being completely blocked at 15°C.

The present paper makes a new and important contribution to our understanding of the factors determining the kinetics of DNA genome infection by phage. Earlier work by the group of the author had established – for phage λ – that the strongly-confined DNA genome of the virus undergoes a first-order ("solid"-to-"fluid") phase transition at a temperature (around 33°C) that depends on local ionic strength. Now, in the present work, it is shown that this phase transition is the "switch" that determines whether the phage, upon binding to its receptor, will eject its DNA immediately – on a time scale of seconds – or do so only over the course of many minutes. Above the transition temperature the "fluid" state of the packaged genome, with relatively low friction associated with the DNA sliding past its neighboring duplex portions, is argued to facilitate fast ejection; conversely, in the low-temperature "solid" state, the high state of stress impedes this dynamical process. Correspondingly, at high multiplicity of infection, as can be guaranteed in vitro by a large excess of phage, the large number of phages bound at the same time will – at temperatures above the solid-fluid phase transition – also eject at the same time, i.e., the ejection will be fast and synchronous. Conversely, if the temperature is below the transition value, the ejection will be slow and asynchronous.

The work uses analyses of a convincing combination of careful and high-quality time-resolved microcalorimetry, time-resolved small-angle-Xray-scattering, and atomic-force-microscopy measurements to conclude that the cross-over from asynchrony to synchrony is indeed determined by the phase transition from "solid" to "fluid" of the phage DNA.

The paper will stimulate important new studies and conclusions concerning the factors involved in the kinetics of phage ejection and its relationship to the choice of lytic versus lysogenic pathways, and all of the reviewers are in agreement about recommending this manuscript for publication without any substantive further work being necessary.

Major points:

The reviewers are also in agreement that – while the manuscript *does* demonstrate convincingly that short-time/synchronous versus longer-time asynchronous injection is correlated with fluid-like versus solid-like organization of the packaged DNA genomes in λ phage – the work does *not* demonstrate correlation of these alternative scenarios with lysogenic versus lytic pathways for the infection. Indeed, all of the experiments reported here are purely in vitro – bacteria-free – ones involving only purified phage and receptor protein under different temperature and salt conditions. As such the work cannot say anything directly about the lysogenic and lytic pathways that ensue upon infection of bacterial hosts by phage. This is not a problem for the paper, however, because the purely in vitro results reported do shed important light on our mechanistic understanding of ejection kinetics as a function of temperature and salt conditions. But we do strongly urge that the Abstract, impact statement, Introduction and conclusions be rewritten to avoid claims about the work contributing directly to the in vivo question of lysogeny versus lysis pathways. The title, as well, should drop the question "Do phages infect in unison?" to read simply "The mobility of packaged phage genome controls ejection dynamics", to reflect the fact that the work presented does not involve any in vivo infection studies (and also to be consistent with the journal policy of avoiding two-part titles).

Minor points

Apart from this point, the reviewers have raised several questions and made several suggestions for correcting and clarifying various minor points throughout the manuscript, to which the authors are requested to respond.

1) There are enough abbreviations that the paper would be easier to read if there were an abbreviations footnote at the beginning.

2) The author should acknowledge that it is not intuitively obvious that when DNA from many phage are delivered simultaneously into the cell the predisposition is for lysogenic infection while when there is asynchronous DNA delivery the infection is lytic. More explicitly, it is argued that in the first situation there is "lack of competition for replication of phage genomes" leading to lysogeny while in the second situation "competitive interactions between phage genomes" leads to lysis. Could the author help the reader better understand this argument?

3) In the Introduction, second paragraph, there is the statement "Tens to hundreds of phage particles adsorb to a single bacterial cell." While this prepares the reader for Figure 1 (where clearly a huge excess of phages has been added to the bacterial host), it is misleading because in the lab the number of phages infecting a cell is determined by the experimenter and can vary by orders of magnitude. For this reason, and because its relevance to what happens in nature is problematic, the figure should probably be dropped altogether.

4) In the fourth-from-last sentence in that same paragraph in the Introduction, "injection events from phage has" should read "injection events from phage have".

5) In the Results section, paragraph starting "To investigate" there is a sentence reading "The second DP peak lasts for ~250 seconds", but none of the Figure 4 curves show results out to 250 sec. Also in this section there's a typo: "encaspdiated" should read "encapsidated".

6) Could the author comment on the significance of the signs of the slopes measured in the calorimetry experiments for enthalpy versus temperature, below and above the transition?

7) The third paragraph of the Conclusions refers to the "genetic switch" and cites the Ptashne book of the same name. This genetic switch refers to the switch that a repressed prophage makes when transitioning to the lytic pathway, upon inactivation of the immunity repressor; it does NOT involve the initial choice made by infecting phage in deciding between the lysogenic and lytic pathways. It is confusing to conflate the two.

8) Can the author say anything about the time scale associated with transition from constrained to fluid states of the packaged genome, upon change in temperature? This would be helpful for allowing the reader to appreciate the connection between the present work and the earlier in vivo studies by Kourilsky in which bacteria are incubated with phages at low temperature and then warmed up to initiate infection and induce choices between lysogenic and lytic options. For example, how does the time needed for the solid-to-fluid transition compare to the time course of the in vivo experiment and to how quickly phage need to eject in order for cooperative (lysogenic) versus competitive (lytic) behavior to ensue?

9) Throughout the manuscript, the bacterium is *Escherichia coli*, abbreviated *E. coli*, and should always be italicized. Also, in the Materials and Methods section: The phage is λ cI857, not cI867.

10) It would be helpful if the caption for each figure included its take-home message rather than just describing its contents, so that the reader doesn't have to spend a lot of time, going back and forth between the figures and the text, trying to figure this out.

---

## [Author Response]

Major points:

*The reviewers are also in agreement that – while the manuscript does demonstrate convincingly that short-time/synchronous versus longer-time asynchronous injection is correlated with fluid-like versus solid-like organization of the packaged DNA genomes in λ phage – the work does not demonstrate correlation of these alternative scenarios with lysogenic versus lytic pathways for the infection. Indeed, all of the experiments reported here are purely* in vitro *– bacteria-free – ones involving only purified phage and receptor protein under different temperature and salt conditions. As such the work cannot say anything directly about the lysogenic and lytic pathways that ensue upon infection of bacterial hosts by phage. This is not a problem for the paper, however, because the purely in vitro results reported do shed important light on our mechanistic understanding of ejection kinetics as a function of temperature and salt conditions. But we do strongly urge that the Abstract, impact statement, Introduction and conclusions be rewritten to avoid claims about the work contributing directly to the in vivo question of lysogeny versus lysis pathways. The title, as well, should drop the question "Do phages infect in unison?" to read simply "The mobility of packaged phage genome controls ejection dynamics", to reflect the fact that the work presented does not involve any in vivo infection studies (and also to be consistent with the journal policy of avoiding two-part titles).*

I agree with reviewers and have now re-written parts of the Abstract, Impact Statement, Introduction and Conclusion sections to clarify the distinction between our in vitro measurements and other observations made in vivo. The title has also been changed.

Minor pointsApart from this point, the reviewers have raised several questions and made several suggestions for correcting and clarifying various minor points throughout the manuscript, to which the authors are requested to respond.1) There are enough abbreviations that the paper would be easier to read if there were an abbreviations footnote at the beginning.

Footnote with abbreviations has now been added on the first page.

2) The author should acknowledge that it is not intuitively obvious that when DNA from many phage are delivered simultaneously into the cell the predisposition is for lysogenic infection while when there is asynchronous DNA delivery the infection is lytic. More explicitly, it is argued that in the first situation there is "lack of competition for replication of phage genomes" leading to lysogeny while in the second situation "competitive interactions between phage genomes" leads to lysis. Could the author help the reader better understand this argument?

I agree, this needed to be further clarified in the text. It has been observed that higher number of simultaneously infecting phages leads to higher lysogenization frequency (Kourilsky P (1973) Lysogenization by bacteriophage lambda. I. Multiple infection and the lysogenic response. Mol Gen Genet 122(2):183-195). Synchronized injections result in immediate presence of DNA from multiple phages in a cell that can be integrated into cell’s chromosome without competition for the progeny. Lysogeny results in the absence of selection, which allows various mutants to integrate and persist in the lysogenic state, leading to diversification of phage genes and adaptation to poor or new growth conditions. On the other hand, if DNA injections from phage are unsynchronized, the first lytic phage to inject its DNA can override the lysogeny decision, so that phages that deliver their genomes with delay will not affect cell’s decision. The early ejected phage will therefore use limited resources to replicate its own progeny. Specifically, it has been shown in ref. (Cortes et a, Biophys J 113, 2110–2120, 2017) that infection delays decrease the chance of lysogeny leading to lytic infection by also lowering CII levels in the cell, where CII is viral protein governing decision-making soon after infection. This has been now more thoroughly explained in the text in both Introduction and Conclusions sections.

3) In the Introduction, second paragraph, there is the statement "Tens to hundreds of phage particles adsorb to a single bacterial cell." While this prepares the reader for Figure 1 (where clearly a huge excess of phages has been added to the bacterial host), it is misleading because in the lab the number of phages infecting a cell is determined by the experimenter and can vary by orders of magnitude. For this reason, and because its relevance to what happens in nature is problematic, the figure should probably be dropped altogether.

Figure 1 has been dropped along with this sentence.

4) In the fourth-from-last sentence in that same paragraph in the Introduction, "injection events from phage has" should read "injection events from phage have".

Has been corrected.

5) In the Results section, paragraph starting "To investigate" there is a sentence reading "The second DP peak lasts for ~250 seconds", but none of the Figure 4 curves show results out to 250 sec. Also in this section there's a typo: "encaspdiated" should read "encapsidated".

The blue ITC titration curve at 22°C in Figure 3A is truncated at 150 s for clarity of presentation, however full ITC titration curve with the same data at 22°C returning to baseline after ~250 is shown in Figure 1 inset. This has been clarified in the text. The typo in word "encapsidated" has been corrected.

6) Could the author comment on the significance of the signs of the slopes measured in the calorimetry experiments for enthalpy versus temperature, below and above the transition?

The slopes in the linear regions of *ΔH_ej_(T)* in Figure 2B, correspond to specific heat capacity *ΔCp(T)* for DNA ejection process [*ΔCp(T)* = *Cp(ejected DNA) – Cp(packaged DNA)*]. Since change in *Cp(ejected DNA)* (i.e., free DNA in solution) is relatively small in the measured temperature interval (22° - 43°C)(Lof et al. 2007), the inversion of *ΔH_ej_(T)* slope at transition temperature (T*~33°C) suggests a change in *Cp(packaged DNA)* value for DNA inside the capsid. This Cp change is in turn associated with transition in ordering and hydration of the packaged genome, which we previously analyzed with ITC (Jeembaeva et al, JMB 2009). This has now been explained in the subsection “Solid-to-fluid like DNA transition in the capsid controls phage ejection dynamics”.

7) The third paragraph of the Conclusions refers to the "genetic switch" and cites the Ptashne book of the same name. This genetic switch refers to the switch that a repressed prophage makes when transitioning to the lytic pathway, upon inactivation of the immunity repressor; it does NOT involve the initial choice made by infecting phage in deciding between the lysogenic and lytic pathways. It is confusing to conflate the two.

This comparison has now been removed and “genetic switch” reference dropped.

8) Can the author say anything about the time scale associated with transition from constrained to fluid states of the packaged genome, upon change in temperature? This would be helpful for allowing the reader to appreciate the connection between the present work and the earlier in vivo studies by Kourilsky in which bacteria are incubated with phages at low temperature and then warmed up to initiate infection and induce choices between lysogenic and lytic options. For example, how does the time needed for the solid-to-fluid transition compare to the time course of the in vivo experiment and to how quickly phage need to eject in order for cooperative (lysogenic) versus competitive (lytic) behavior to ensue?

These are very interesting and relevant points. The ITC instrument used in this study for DNA ejection dynamics, requires at least 15 minutes to equilibrate at a given temperature. This time interval is sufficient for DNA transition to occur. We have also previously observed with time-resolved SAXS measurement that 10 min incubation is also sufficient for intra-capsid DNA transition to take place. In order to investigate the effect of even shorter time scales on DNA transition dynamics, a different set of measurements involving other techniques would need to be designed due to technical limitations of temperature equilibration of sample cell in ITC and SAXS systems. Regardless, Kourilsky experiments involve at least 10 min long incubations (or longer), which indeed provides sufficient time for intracapsid DNA transition to occur.

To address the second part of the question: “how quickly phage need to eject in order for cooperative (lysogenic) versus competitive (lytic) behavior to ensue?” As explained in the manuscript, the translocation time for DNA from a single phage is very rapid and occurs on a time scale of seconds (approximately 10 seconds). It is however the ejection delays from multiple phages infecting a cell that lead to higher lytic infection probability. It was shown (Cortes et a, Biophys J 113, 2110– 2120, 2017) that the cutoff time where delayed phage DNA injection no longer affects the lytic/lysogenic cell decision is on the time scale of few minutes to tens of minutes depending on the MOI (the higher MOI the shorter is the cutoff time). Comparable timescales for overall ejection delays are observed in our measurements which suggests that DNA ejection delays due to solid- to-fluid like DNA transition will affect the lytic/lysogenic decision. However, while it has been shown that probability of lysogeny depends inversely on phage infection delays, the infection timings alone cannot be used to exactly predict the outcome of either lysogenic or lytic behavior since other factors such as cell volume and host-cell physiology have also been shown to play a role (Francois St-Pierre et al PNAS, 2008, vol. 105, no. 52). This has now been further clarified in the Conclusions section.

9) Throughout the manuscript, the bacterium is Escherichia coli, abbreviated E. coli, and should always be italicized. Also, in the Materials and Methods section: The phage is λ cI857, not cI867.

This has been changed.

10) It would be helpful if the caption for each figure included its take-home message rather than just describing its contents, so that the reader doesn't have to spend a lot of time, going back and forth between the figures and the text, trying to figure this out.

Take home messages have now been added to every figure caption.